# A Participatory, Needs-Based Approach to Breastfeeding Training for Confinement Centres

**DOI:** 10.3390/ijerph191710914

**Published:** 2022-09-01

**Authors:** Siew Cheng Foong, Wai Cheng Foong, May Loong Tan, Jacqueline J. Ho, Amal Omer-Salim

**Affiliations:** 1Department of Paediatrics, RCSI & UCD Malaysia Campus (Formerly Penang Medical College), George Town 10450, Malaysia; 2World Alliance for Breastfeeding Action, George Town 10850, Malaysia

**Keywords:** breastfeeding training, confinement centre, participatory approach, needs-based approach, dialogue, Chinese, Malaysia, support, cultural factors, traditional practices

## Abstract

With a focus on traditional practices rather than evidence-based practices, breastfeeding support is sub-optimal in confinement centres (CCs). We used a participatory, needs-based approach to develop a training module for CC staff adopting Kern’s six-step approach as our conceptual framework. Of 46 identified CCs, 25 accepted our invitation to a dialogue aimed at establishing relationships and understanding their needs. An interactive training workshop was developed from the dialogue’s findings. The workshop, attended by 32 CCs (101 participants), was conducted four times over a four-month period. Questions raised by the participants reflected deficits in understanding breastfeeding concepts and erroneous cultural beliefs. Correct answers rose from 20% pre-test to 51% post-test. Post-workshop feedback showed that participants appreciated the safe environment to ask questions, raise concerns and correct misconceptions. An interview conducted 14 months later showed that while some CCs improved breastfeeding support, others made no change due to conflict between breastfeeding and traditional postnatal practices, which was aggravated by a lack of support due to the COVID-19 pandemic. A participatory approach established a trustful learning environment, helping CCs appreciate the value of learning and adopting new concepts. However, cultural perceptions take time to change, hence continuous training and support are vital for sustained changes.

## 1. Introduction

For generations, mothers of Chinese ethnicity in Asia adhere to traditional postpartum practices for 30 days after childbirth, commonly known as the confinement period. During this time, a new mother is ‘confined’ to her home while she observes practices aimed at restoring her well-being. Failure to do so is believed to be detrimental to the mother’s health [1,2].

Malaysia is a multi-ethnic country, and the ethnic Chinese form one of the three major ethnic groups. They make up 39% of the population of Penang, a state northwest of Malaysia. Despite being known as a progressive and modern population, traditional confinement practices after childbirth remain inherently important to Chinese mothers in Penang. Several of these practices are based on the ancient Chinese philosophy that emphasises the importance of maintaining the balance between the ‘Yin’ and the ‘Yang’. The ‘Yin’ and ‘Yang’ are opposites in nature and the attributes of all things and phenomena in the universe. ‘Yin’ generally refers to ‘cold’, whereas ‘Yang’ generally refers to ‘warmth’. Childbirth is believed to compromise the ‘Yang’, hence practices which help the mother restore the ‘Yang’, such as getting enough rest and keeping ‘warm’ by consuming food that are ‘heaty’, are emphasised. There is also an emphasis on avoiding wind (draughts) and avoiding showers or baths during the confinement period. These practices can affect breastfeeding practices negatively; for example, when the mother tries to have more ‘rest’ at the expense of limiting the frequency of breastfeeds [1,3,4,5]. For more information, please refer to our previous papers [6,7].

In the past, traditional confinement was practiced in the mother’s own home with the help of a hired postpartum carer, known locally as the ‘confinement lady’. Besides the traditional care from the hired postpartum carer, these mothers also get home visits from government health services. However, a new trend has been the development of postpartum centres, known locally as ‘confinement centres’ (CC), to help mothers observe these traditional confinement routines outside of their homes.

All CCs in Malaysia are private establishments set up as a business model and are unregulated by health authorities in Malaysia. They are usually residential or commercial properties that have been converted to house mothers and their newborns. The Malaysian health ministry does not formally recognise these establishments, hence mothers staying in CCs do not get the routine postnatal visits from the healthcare staff that mothers at home get. Therefore, routine postnatal care is given by CC staff who may be trained healthcare staff, such as nurses or midwives, but may also be women who are considered as experts in the traditional cultural confinement and diet requirements but without any formal healthcare training. These traditional carers would have obtained their skills through experience rather than formal training [4,7].

In a study done in 2018 documenting mothers’ experience during their stay in CCs in Penang, we found gaps in breastfeeding support [6,7]. As breastfeeding is key to the survival, health and wellbeing of babies and women [8], proper breastfeeding support is necessary [9,10,11]. Training providers of breastfeeding support is needed irrespective of whether they are healthcare workers or anyone looked upon by the mothers as a support person, such as the CC staff [9]. Initiatives in South Africa, Nigeria and Bangladesh have shown that training community health workers and peer counsellors is effective [12,13,14]. While CC staff were neither healthcare workers nor peer counsellors, the principle of training people who support breastfeeding women could be applied here. A systematic review on breastfeeding education and training shows a lack of knowledge about how to deliver breastfeeding training programs effectively in diverse settings [9].

From the gaps identified in the 2018 study, we conducted a participatory, needs-based workshop for CC staff in Penang. We used Kern’s [15] six-step approach as the conceptual framework to guide the development of an effective training module for CC staff. The six steps were: problem identification, needs assessment of the learners, educational goals and objectives, instructional strategies, implementation, evaluation and feedback. This framework, derived from the systems theory, helped us visualise different perspectives and identify a broader range of available solutions [16].

The aim of this paper is to describe our experiences with the development and implementation of the workshop, highlight enablers and challenges in the process and share lessons learned. A study on the impact of the workshops on breastfeeding and the effect of hygiene on mothers’ experience in CCs has been completed and will be published separately.

## 2. Materials and Methods

### 2.1. Problem Identification (Kern’s Step 1)

This had been done via our 2018 study which revealed gaps in how breastfeeding was supported in CCs. This included a lack of support for mothers to room-in with their babies or directly breastfeed their babies [7].

### 2.2. Needs Assessment: Dialogue with CCs (Kern’s Step 2)

Prior to the workshop, we organised a dialogue session with all CCs in Penang to understand their concerns and needs. A list of CCs operating in the state of Penang was compiled through searching the internet and social media for published websites or pages. The contact details were obtained from these sites and invitations were then sent via letters, emails and/or text messages.

During the dialogue, participants were divided in small groups with an assigned moderator and a rapporteur. The moderators were either paediatricians or lactation nurses while the rapporteurs were recent medical graduates.

Within each group, the moderator asked a set of questions (Appendix A), mainly to understand the group’s perspectives and issues related to breastfeeding when caring for postpartum mothers and their newborns, as well as their interest in and need for training. The group rapporteur documented all responses. At the end of the session, the responses were collated and analysed for recurring themes.

### 2.3. Educational Goals (Kern’s Step 3)

Our goal was to develop knowledge, behaviours and skills among CC staff to be able to support breastfeeding in their respective centres.

### 2.4. Instructional Strategies: Development of the Training Package (Kern’s Step 4)

In order to achieve our goals, we worked together with paediatricians, lactation nurses and nutritionists to develop a training package.

Issues brought up by mothers who participated in the 2018 study [6,7] as well as themes from the CC dialogue were reviewed to identify the breastfeeding support gaps and learning needs. As some of the learning needs identified were related to use of traditional treatment, advice was also sought from trained traditional Chinese medicine practitioners to align the training package with traditional cultural beliefs.

Based on the findings, and using our educational background as academicians in a medical school, we designed a one-day workshop based on the framework of the World Health Organization (WHO) 20-h workshop [17]. We applied the constructivist view that all learners desire to find meaning in what they learn rather than just accumulate information—we used this view to modify and prioritise the list of topics based on the main needs that emerged from the dialogue [18]. We wanted the participants to appreciate the value of the course and hence, be motivated to learn [19].

In order to ensure active participation, we paid special attention to the method of delivery of the topics in our planning. The workshop programme was designed to allow participants to engage in active learning with elements of experiential learning. Ideas and procedures that can be visualised are also more likely to be remembered [20]. Therefore, the workshop included various hands-on activities and role plays so that participants could understand the real-life applicability of what they were learning. We also included several small group discussions with case scenarios that provided opportunities for participants to articulate their prior knowledge and understanding of the issues.

To encourage participants to reflect on what they have been doing to support breastfeeding and what steps they would take to improve breastfeeding support in their CC after the workshop, we designed a self-reflection form for participants to fill out. This was achieved by adapting the 1992 Baby Friendly Hospital Initiative (BFHI)’s ten steps [21] for the context of a CC. For example, Step 1, “Have a written breastfeeding policy that is routinely communicated to all healthcare staff”, from the BFHI’s ten steps became “Does your centre have a breastfeeding policy? Is this a written policy? Are there any free or low-cost formula supplies, or any other promotional material from formula milk companies, e.g., calendars supplied by milk companies with their logos?” (see Appendix A).

### 2.5. Implementation: Delivery of the Training Package (Kern’s Step 5)

All CCs in Penang were invited to participate in the one-day interactive workshop. To ensure that small group discussions were interactive, we restricted the number of participants in the workshop. Therefore, each workshop was limited to a maximum of 36 participants. The workshop participants were divided into small groups of maximum six people per small group. Each small group had a facilitator to guide the discussion. As the response and demand for the workshop was very good, this meant we had to repeat the workshop four times between June and October 2019. Due to the variation in the participants’ language preference, three of the workshops were conducted mainly in Mandarin Chinese, while one was conducted mainly in the Malay language mixed with English.

### 2.6. Evaluation and Feedback: Post-Training Feedback and Evaluation (Kern’s Step 6)

We evaluated the workshop by two means, a pre- and post-test as well as individual feedback from participants.

At the beginning of each workshop, participants completed a pre-test consisting of multiple-choice questions on areas related to breastfeeding and hygiene that would be covered during the workshop (Appendix A). The same test was administered as a post-test at the end of the workshop. In addition, all participants were asked to complete the self-reflection form about how they were currently supporting breastfeeding in their respective centres and what they would like to change after the workshop (Appendix A). We also invited them to fill up a feedback form (Appendix A) to let us know how they found the workshop and solicit their suggestions on what could be done to improve it.

Fourteen months after the workshop, telephone interviews were conducted with each CC that participated. The aim was to evaluate changes in practices by CCs. All CC managers who had sent their staff to one of the workshops were contacted by a trained interviewer. After obtaining informed consent, the participant was interviewed based on a specially designed standard interview format. Questions included details of any new breastfeeding support they had introduced since the workshop, any difficulties they faced doing this, anything they would have liked to do but found too challenging, any particular training they think might help them further and if our earlier workshop should be modified accordingly.

## 3. Results

### 3.1. Assessing Needs: Dialogue with CCs

A total of 25 out of 46 identified CCs responded to the invitation and sent a total of 36 managers or senior administrative staff to the dialogue session in January 2019. All participants were divided into six groups of six participants each. We were surprised at the degree of enthusiasm and engagement of the participants.

From the small group discussions, it was clear that all CCs were providing breastfeeding support but at varying degrees. These ranged from measures as simple as encouraging the mothers in their care to breastfeed to providing specific education on breastfeeding. However, they faced difficulties in providing the support. The main barriers were CC staff lacking the knowledge and skills necessary to support a breastfeeding mother, mothers’ lack of interest in breastfeeding (especially direct breastfeeding) and resistance towards breastfeeding by the grandparents of babies who visited the CCs. The CCs were very keen to attend training and suggested that training should be officially certified. Examples of topics they would like to be included in the workshop were how to help a mother with breastfeeding positioning and attachment, management of common breastfeeding problems, handling expressed breastmilk, methods to increase milk production including the role of foods, effective communication with mothers and family members and common non-breastfeeding related problems in the newborn (jaundice, colic, umbilical cord care etc.). The CCs also suggested the types of support they themselves would need. These included scheduled visits by healthcare professionals or lactation consultants to assess mothers and babies, educational resources (such as talks and training), a list of local certified lactation consultants or breastfeeding support groups, a breastfeeding helpline, breastfeeding-friendly accreditation for CCs and how to kick-start a CC organization that would encourage all CCs to self-regulate and support each other.

### 3.2. Development of a Practical Training Package That Addressed Culturally Specific Issues

From the above findings, we developed an interactive one-day workshop which focused on breastfeeding and hygiene with consideration given to traditional cultural beliefs and practices including the postnatal prohibitions and requirements normally observed during the confinement period.

Examples of topics covered in the training sessions include how breastfeeding is more than just breastmilk production, hence the importance of direct breastfeeding and not just expressed breastmilk feeding; basic physiology about how a baby gets milk from the breasts, hence the importance of rooming-in; breastfeeding positioning and attachment; managing common breastfeeding problems; hand hygiene and handling of expressed breast milk; and relevant communication skills and best practices to support breastfeeding (see Appendix A).

### 3.3. Delivery of the Training Package

A total of 101 participants from 32 different CCs attended the workshops. Of these, 28% were managers or someone with a senior administrative role in the CC, and the rest were staff who took care of the mothers and babies. Three CC managers returned for all three workshops to support the staff who attended. Of note, all of the 25 CCs who came for the earlier dialogue sent at least one participant to the workshop. In addition, another seven CCs also sent participants to our workshop when we sent out the call for participation.

The workshop generated a lot of discussion and questions from the participants. The questions and discussion points raised by the participants often reflected that they had not fully understood the concepts of breastfeeding and were confused by misinformation and erroneous cultural beliefs. For example, two questions that were often asked were, “Should a mother stop breastfeeding if the baby is jaundiced?” and “Should breastfeeding mothers avoid ‘cooling’ food like cabbages to avoid infantile colic?”. The lack of of government community nurses to make routine post-natal visits to CCs was raised again during the workshop, similar to how it was raised during the initial dialogue.

### 3.4. Self-Reflection to Consolidate Learning

During the self-reflection exercise, participants reflected on what they were currently doing and what they could improve upon after attending the workshop. Most of them committed to displaying a written breastfeeding policy at their CC and several mentioned that they would try to increase breastfeeding awareness and help mothers breastfeed with their newly acquired skills.

### 3.5. Evaluation of the Workshop

#### 3.5.1. Pre- and Post-Tests

Pre- and post- tests were only administered for the second, third and fourth workshops (*n* = 79). However, results of the pre-tests were not included for the 16 participants who arrived late for the workshop. Results were therefore available for 63 of the total 101 participants from the four workshops. It is to be noted that a small number of these participants had come for the workshop three times, hence the same participant could have submitted the pre- and post-test three times.

There was an increase in participants who answered all the questions correctly, from 12/63 (20%) to 32/63 (51%), after the workshop. The most marked improvements were seen for the questions on the following: indications that the baby has fed adequately, causes of poor milk supply, how to improve breast milk supply and what to do if the mother has nipple pain during breastfeeding. Questions participants continued to respond to incorrectly were whether or not an exclusively breastfed baby should be given additional water, whether jaundiced babies could continue breastfeeding and how long a baby should be exclusively breastfed.

#### 3.5.2. Post-Workshop Evaluation

Immediate feedback revealed that participants found the workshop enlightening and appreciated being able to freely ask questions, have opportunities to correct misinformation and raise concerns regarding issues with the authorities.

### 3.6. Follow-Up Telephone Interviews

Of the 32 CCs we contacted, sixteen completed the interview, five were not contactable, three informed us that they had closed or were intending to close down following repeated lockdown orders due to the COVID-19 pandemic, eight declined to participate due to a change in ownership or lack of interest. We were unable to determine if there were any systematic differences between CCs who completed the interview and those that were not contactable.

From the interview, we found that about half of the centres that previously provided minimal breastfeeding support (in particular direct breastfeeding support) had started to do so after the workshop. However, among those that previously gave free samples of milk formula to their clients, none stopped doing so after the workshop. In addition, despite noting that CCs had made the resolution to have a breastfeeding policy in their centre after the workshop, only one of them had actually done so (refer to Table A1).

During the interview, the CCs also reported ongoing challenges faced. These included poor awareness among clients about the importance of rooming-in and direct breastfeeding, thus making it difficult for mothers to accept this concept. Therefore, the practice of rooming-in and getting mothers to directly breastfeed instead of only feeding with expressed breast milk was found to be difficult to initiate and sustain. Many CCs continued to have difficulties in managing breastfeeding problems such as engorgement. In fact, the pandemic had increased these difficulties because no lactation consultants were going into their premises during the COVID-19 pandemic. Three centres reported that they wanted to organise breastfeeding education for their clients but faced several challenges in doing so. Among reasons cited was poor demand and frequent turnover of mothers and staff. Two CCs mentioned that they wanted to encourage their clients to continue breastfeeding after leaving their centre but did not have the resources to do so. All of them said that they needed us to continue conducting training or refresher workshops on a long-term basis. When asked how they would like to modify our earlier workshop, five CCs asked if we could help them conduct training courses for their clients and one asked if we could include more information on culture-based practices such as whether or not a mother could bathe during the confinement period. Three CCs informed us that they had been struggling to keep their business going during the COVID-19 pandemic and were likely to cease operations.

## 4. Discussion

CCs are an important resource for improving the care of mothers and newborns in our community. Chinese mothers in Malaysia are deeply rooted in traditional practices, hence CCs meet the demand for traditional practices. At the same time, CCs can be used to improve coverage of important healthcare practices including breastfeeding. However, their effectiveness is limited unless these CC staff have proper training and good support from people with relevant skills.

We have described the process and results of conducting the workshops targeting CCs. There were many challenges faced throughout the process and we learned many lessons. We share the key lessons that may be applied when planning similar training programmes in the future.

**Lesson** **1.**
*Gaining trust lays the foundation for a successful training programme.*


Gaining the trust of the CC operators was one of our early challenges. We were aware of the existing tension between CCs and health authorities up to this point of time. CCs were reportedly suspicious about the actions of health authorities because of past experience [22]. Therefore, when we contacted the CCs to attend the dialogue, we were not sure how many would actually respond. In addition, we were uncertain about how they, as business competitors, would be able to work together. However, the overwhelming positive response to the initial dialogue and subsequent training showed that our initial concerns were unfounded. CCs wanted to play a role in supporting breastfeeding in their centres but had never been approached before. The relationship and social bonding built through the initial dialogue became the foundation of the subsequent workshops. During the course of the workshop, we gave the participants numerous opportunities to interact and ask questions in a safe environment and this led them to begin asking questions including those that were deemed sensitive. It was here that we could address some of these sensitivities as well as deal with some of the misconceptions they may have had. The participatory design allowed them the opportunity for collaborative dialogue and provided them with a platform for sharing their views [23]. Social educational theories recognise that the interaction and sharing of knowledge between participants are essential for effective learning [19,24,25]. Participants’ learning is enhanced when given the opportunity to articulate their thinking to peers or themselves [26]. More importantly, we found that by carefully developing trustful relationships with the CCs, they became convinced of our good intentions, and this in turn helped them appreciate the value of learning new concepts and methods to support mothers and newborns in their centres.

**Lesson** **2.**
*Cultural perceptions take time to change.*


Although we took care to design the content and delivery style of the workshop based on the initial dialogue with CC managers, it became clear that the CC staff were mostly concerned about cultural perceptions of breastfeeding practice. In all of the four workshops, questions about cultural practices kept emerging, for example, the question about whether food deemed to be ‘cooling’ would harm the baby if the mother breastfed while taking them. They also wanted to know if it was safe for the baby if a breastfeeding mother consumed various traditional herbs or tonics. In accordance with traditional thinking, many had the perception that mothers must not consume food that was yellow or orange in colour (e.g., carrots, pumpkin) to avoid the baby from becoming jaundiced. There was also the question of whether breastmilk was ‘heaty’ and caused a rash and if this ‘heat’ would be exacerbated by not giving the baby water to drink. It was interesting to note that these questions did not only come from lay participants, but also trained nurses, thus proving how much influence traditional cultural beliefs have on their daily practice. A significant amount of time was dedicated during the workshops to address these concerns. However, during the post-workshop telephone interview with CCs, issues related to culture-based practices were still apparent, demonstrating that changes in cultural perceptions take time. Our approach to addressing these issues was to explain the rationale behind some of these beliefs to help them understand how situations have evolved over time and how to align them with science using pragmatic examples that could be appreciated by CCs. This is in line with a study by Im [23] which highlighted the importance of having skills to address cultural reservations to healthcare practices as some of the practices advocated by healthcare professionals are essentially a new culture to them. Aligning scientific knowledge with cultural practices may help CCs be more willing to change.

**Lesson** **3.***Training can empower and bring about champions*.

Training CC staff and empowering them with evidence-based knowledge is vital and could help them be agents of change to improve breastfeeding practices [27,28,29]. Community-based health workshops have been shown to be beneficial in building capacity for health promotion [23]. Indeed, at the end of our workshops, many of the participants felt that they have a worthwhile role to play in promoting breastfeeding. In fact, we were very encouraged to find that there were three participants from different CCs who were very inspired and committed to move forward and make changes to their practice. They also engaged a few other CC managers to join our subsequent workshops and had plans to form a confinement centre association. Their goal was to have the association set standards for CCs to adhere to.

The legitimation of their role could also increase the health capital of mothers through experiential knowledge sharing, social support and connectivity [30], and bring them closer as a community healthcare system. In Asia, where tradition and culture have a big role in influencing mothers’ decisions, these champions for breastfeeding would provide better peer support. Grandmothers have a strong influence over mothers because they are highly respected. Mothers were more likely to follow their family members’ instructions regarding postpartum practices even if the advice given conflicted with advice from healthcare professionals [31,32]. This is worrying because their knowledge was often inadequate or out of date. Many of the CC staff are akin to grandmothers in that they are the bearers of traditional practices and their knowledge about postpartum care could be limited to information from like-minded peers rather than scientific or evidence-based sources [30]. Therefore, the training of CC staff empowered them with the right skills to be champions for breastfeeding among the mothers who look up to them.

**Lesson** **4.***Embedding reflection into the programme may help behavioural change*.

Reflection is an integral part of workshops. Therefore, towards the end of the workshop, one of the activities was participant reflection on what they have been doing well to support breastfeeding, then to think about how they would use the new information obtained from the workshop to improve breastfeeding support in their CC after the workshop. Research has shown that the acquisition of information alone is unlikely to result in change unless accompanied by an opportunity to use that information [33,34].

In addition, the pre-test and small group discussions evoked our participants’ prior knowledge so that they could either reconstruct or build upon it during the workshop. This aligns with the learning principles that underpin the power of prior knowledge in how adult learners process and understand new information and helps them be more open to the prospect of change [19,20]. It also enabled us to recognise specific misconceptions and problems they face so that we could address them more effectively.

**Lesson** **5.***Mentoring is vital*.

Even when the pandemic first hit, the few CCs who were championing for change initially persevered to move ahead to form an association. However, the group was small, and their voice was not loud enough to get more CCs to join them. They also relied on continued support from us and continued to contact us for help with various breastfeeding issues. Unfortunately, apart from our small team, there was no one else from any other organization who could provide this support. Due to our own commitments, we were unable to continue providing the intensive support we had hoped to provide, and this led to the stalling of the establishment of a CC association which was expected to provide self-regulation and self-support for the CC community. Being unable to meet this need might have affected the trust we had carefully built, which could explain why some of them were not as approachable when we tried to re-connect with them to find out how they were doing 14 months later. Results from the follow up telephone interview showed that changes in practice were small despite the positive feedback and improved post-test scores. Almost all of them asked for ongoing refresher courses for their staff. As in previous studies, training alone is not likely to change practice [35]. Our one-day training was not adequate to provide the skills needed to sustain improved practices but did result in the establishment of relationships that led to mentoring opportunities. Practices and behaviours are difficult to change unless there is acceptance and continued support [35]. Continued mentoring would also improve motivation and retention and allow them to establish their role in the community [36,37].

**Lesson** **6.***Creating demand is essential*.

CCs who completed the phone interview lamented about how difficult it was for them to change practices because there was no demand for such change from their clients. For example, rooming-in was an important measure we advocated for during the workshop to help with breastfeeding. Unfortunately, we were informed that many CCs could not implement this because their clients objected to having to room-in with their baby. They preferred that the CC staff take care of the baby while they rested—rest being culturally very important to Chinese post-partum mothers [4,7,38,39]. There is a fear that their clients may spread negative information about their centres on social media. There have been unverified reports of negative social media messages such as “lazy staff” and “forced to room-in” because mothers were asked to care for their own babies at the CC. Similarly, due to the expectations of clients and pressure to stay competitive, a lot of CCs were still giving out infant formula samples.

Unless mothers understand the importance of rooming-in with their babies, or the harms that can arise from infant formula samples, they would not be able to accept these new practices of the CCs [40]. Therefore, it is vital to create awareness so that there will be public demand for these measures.

### Limitations

The main limitation is that the repeated, nationwide, mandated lockdowns hindered the CC’s ability to provide postnatal care, hence it was difficult for us to observe any potential effects of the workshop. Furthermore, there was a shift in focus for our team of healthcare professionals involved in this project. Priority had to be given to dealing with the pandemic, limiting our ability to fulfil our planned activities to provide post-workshop support to the participants.

CCs also struggled to continue operations because many of their clients chose to cancel their stay out of fear of infections. Due to the repeated nationwide lockdowns imposed by the government to curb the spread of COVID-19 infections and the lack of clarity in standard operating procedures, many CCs were unsure of whether they could operate. Some chose to stop operating rather than take the risk of being found unlawfully operating. In the end, many CCs were forced to cease operations for financial reasons. The development of their newly acquired skills was no longer their priority as they had to divert their attention to the many issues related to the pandemic and work on keeping their business alive. Another limitation is the low response rate in both the pre- and post-test and phone interviews.

Despite these limitations, we are able to describe the process of dialogue, training and immediate outcomes, although the effects 14 months later are still uncertain.

## 5. Conclusions

The participatory approach allowed CCs an opportunity for collaborative dialogue and provided them with a platform to share their views which, in turn, helped inform the content and delivery of our workshop. This has a unique strength in empowering CCs with resources and skills to offer community-based health education.

Establishing a safe learning environment and trustful relationship are important factors to be able to fully understand challenges faced by CCs and be able to correct misinformation especially when it involves cultural practices. It also helps draw out champions from within the community who could then be the agents of change to improve the health of post-partum mothers and newborns in CCs. However, continued support and mentoring are essential to allow actual sustained change in practice. There must also be public awareness about the need to support breastfeeding and encourage demand for such services.

Our model for training received positive feedback and showed initial improvements in our participants’ knowledge. However, we were unable to observe any impact beyond that. An in-depth qualitative study is warranted to further refine the training framework based on culturally appropriate and universal elements [23].

## Data Availability

The data presented in this study are available as Appendix A.

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
