# Peer review of "A Participatory, Needs-Based Approach to Breastfeeding Training for Confinement Centres"

_ijerph, 2022, doi:10.3390/ijerph191710914_

Round 1
Reviewer 1 Report
-Results are not displayed accurately.
-Needs more statistical tests (differences between before and after): The study need statisticals analysis of the differencess between the pre- and post-programme. As the authours in this study showed only post -result , and this didi not fit with the design of study.
Author Response
Thank you for your comments. Please see the attachment for our response. Thank you.

Reviewer 2 Report
The manuscript entitled “A participatory, needs-based approach to breastfeeding training for confinement centres” by Foong, S.C. et al. presents the results derived from a participatory needs-based approach to develop a training module on breastfeeding support for the staff of confinement centres in Penang (Malaysia). The manuscript is well-written and structured and the authors clearly transmit their main concerns in promoting breastfeeding in confinement centres. However, after reading the manuscript, there are some issues that in my opinion need to be addressed in order to be suitable for publication.
· The authors state that they used the Kern’s six-step approach as conceptual framework. However, to what extent this was apply remains unclear. Can the needs assessment of the learners be listed or clearly stated? Which were the educational goals of the workshop? (the general purpose can be inferred, but concise objectives are lacking)
· The authors do not mention the importance of milk composition in the benefits of breastfeeding and its changes along lactation, during the day, within a feed, between mothers, etc. Was this considered in the training programme?
· Have the authors information of how many mothers attended on average these centres per year?
· I would suggest to include a list of the cultural beliefs about breastfeeding practices, since these are expected to be different from other cultures and helps the reader to understand what difficulties face the authors.
· Lines 153 and 187. The authors state that 25 CCs accepted their invitation to an initial dialogue, but in total 32 CCs participated. It is not clear why the numbers differ.
· Lines 208-211. Why the pre- and post- tests where not administered in the first and last workshop?
· Line 140. The feedback form is not available for the readers. Please, include it.
· Lines 198-199. Table A1 is referred, but no connection with the text arises.
· Line 414. Do the authors mean 14 months instead of 12?
· Additional comments:
o Some sub-sections are numbered, but others don’t (i.e. Materials and Methods). Please, be consistent or follow journal guidelines.
o Line 118. Introduce acronym (BFHI) used afterwards in line 120.
o Line 336. Verb repeated in two different tenses.
o Name Appendix 2-4 (headings) in the supplementary information.
Author Response
Thank you for your comments. Please see the attachment for our responses. Thank you.

Round 2
Reviewer 1 Report
The previous comments have not been modified.
Author Response
It seems that Peer Reviewer 1 would still like us to perform a statistical analysis, so the research team have once more reviewed this. In the end, with due respects to the reviewer, we still strongly feel that statistical analysis should not be done. Perhaps we did not word this clearly in our earlier response to the Peer Reviewer hence we will try to explain our reasons and concerns more clearly here. This is not a quantitative paper and is not to designed to inform whether our workshop was effective. The objective of the paper was NOT to show the differences before and after the workshop. Instead, as per what we had written in Line 85-87 of the submitted manuscript, we have stated that the objective of the paper was to describe what we had done since we found gaps in how breastfeeding was supported in confinement centres in our 2018 study. Using Kern’s 6-step method as our conceptual framework, our aim was to add to the body of knowledge about how a participatory method was used to build a relationship with the confinement centres, develop the programme content, and share the lessons learnt in the process. As we were not testing a hypothesis in this paper, sample size calculations were not done. Our sample size was very small, rendering statistical tests meaningless as positive results could be simply due to chance and negative results could simply be because the sample size had been inadequate. In other words, having statistical tests will not bring any added value, hence our earlier decision not to include them in this paper still stands to prevent misuse of statistics.
However, we acknowledge that perhaps the reviewer had picked up something that we had overlooked. If so, we would appreciate if the reviewer could kindly point out why and what statistical tests are needed to add value to our paper (bearing in mind that this is not a quantitative study, and that we are NOT evaluating the pre- and post- workshop results). Once we understand the rationale behind how and what statistical tests will add value to this paper, we will be happy to re-consider our decision.
It might be useful to note that upon the completion of the 4 workshops described in the paper, we had indeed done a follow-up study subsequently to see the impact and how effective our workshops had been on mothers’ breastfeeding and hygiene experiences. In that study done in 2020, sample size calculations were done, and we would be using statistical tests to compare our results pre and post workshop. We mentioned in our manuscript that this follow-up study had been completed and the results will be published as a separate paper.
We hope this explains the situation more clearly and we hope you will give our concerns due consideration. Thank you.
(You can also see the attachment.)
